Recursion to food plants by free-ranging Bornean elephant

English Megan 1 megs.english@gmail.com
Gillespie Graeme 2
Goossens Benoit 3 4 5
Ismail Sulaiman 6
Ancrenaz Marc 6
Linklater Wayne 1 7
1 Centre for Biodiversity and Restoration Ecology, School of Biological Sciences, Victoria University of Wellington , New Zealand
2 Department of Zoology, University of Melbourne , Parkville, Victoria , Australia
3 Danau Girang Field Centre, c/o Sabah Wildlife Department , Kota Kinabalu, Sabah , Malaysia
4 Organisms and Environment Division School of Biosciences, Cardiff University , Cardiff , UK
5 Sabah Wildlife Department , Kota Kinabalu, Sabah , Malaysia
6 HUTAN Elephant Conservation Unit and Kinabatangan Orangutan Conservation Project , Sukau Sabah , Malaysia
7 Centre for African Conservation Ecology, Nelson Mandela Metropolitan University , Port Elizabeth , South Africa
Cowling Richard
Electronic publication date: 2015 Aug 4
Publication date: 2015
Volume: 3
Electronic Location ID: e1030
Received 2015 Mar 4; Accepted 2015 May 26
Copyright: © 2015 English et al.
Copyright year: 2015
Copyright holder: English et al.
License: This is an open access article distributed under the terms of the Creative Commons Attribution License, which permits unrestricted use, distribution, reproduction and adaptation in any medium and for any purpose provided that it is properly attributed. For attribution, the original author(s), title, publication source (PeerJ) and either DOI or URL of the article must be cited.
License URL: https://creativecommons.org/licenses/by/4.0/

Keywords: Recursion, Elephant feeding behaviour, Herbivore-resource relationships, Herbivore ecology, Re-browsing

Funding: ZoosVictoria, Australia Victoria University of Wellington This project was funded by ZoosVictoria, Australia, and research scholarships provided by Victoria University of Wellington, New Zealand. The funders had no role in study design, data collection and analysis, decision to publish, or preparation of the manuscript.

==============================
Plant recovery rates after herbivory are thought to be a key factor driving recursion by herbivores to sites and plants to optimise resource-use but have not been investigated as an explanation for recursion in large herbivores. We investigated the relationship between plant recovery and recursion by elephants (Elephas maximus borneensis) in the Lower Kinabatangan Wildlife Sanctuary, Sabah. We identified 182 recently eaten food plants, from 30 species, along 14 × 50 m transects and measured their recovery growth each month over nine months or until they were re-browsed by elephants. The monthly growth in leaf and branch or shoot length for each plant was used to calculate the time required (months) for each species to recover to its pre-eaten length. Elephant returned to all but two transects with 10 eaten plants, a further 26 plants died leaving 146 plants that could be re-eaten. Recursion occurred to 58% of all plants and 12 of the 30 species. Seventy-seven percent of the re-eaten plants were grasses. Recovery times to all plants varied from two to twenty months depending on the species. Recursion to all grasses coincided with plant recovery whereas recursion to most browsed plants occurred four to twelve months before they had recovered to their previous length. The small sample size of many browsed plants that received recursion and uneven plant species distribution across transects limits our ability to generalise for most browsed species but a prominent pattern in plant-scale recursion did emerge. Plant recovery time was a good predictor of time to recursion but varied as a function of growth form (grass, ginger, palm, liana and woody) and differences between sites. Time to plant recursion coincided with plant recovery time for the elephant’s preferred food, grasses, and perhaps also gingers, but not the other browsed species. Elephants are bulk feeders so it is likely that they time their returns to bulk feed on these grass species when quantities have recovered sufficiently to meet their intake requirements. The implications for habitat and elephant management are discussed.

Introduction

Recursion by wild herbivores is the repeated use of the same sites or finer-scale reuse of resources, such as individual plants, within a site over time. Importantly, recursion by wild herbivores to previously browsed or grazed sites and plants is thought to facilitate plant productivity (re-growth) and its consumption at stages of highest productivity (McNaughton, 1985; Gordon & Lindsay, 1990; English et al., 2014a). Recursion may also accelerate nutrient cycling at sites (Gordon & Lindsay, 1990; McNaughton, Banyikwa & McNaughton, 1997) and so maintain them as nutrient hotspots (Winnie, Cross & Getz, 2008). Thus, recursion is thought to trigger and maintain the positive feedback between large herbivore feeding, and vegetation regeneration and palatability (McNaughton, Banyikwa & McNaughton, 1997). Although recursion is consistent in these ways with optimal-foraging theory and strategies, and assumed to be ubiquitous (McNaughton, 1985), it has only rarely and recently been investigated amongst wild herbivores.

Recursion has been described at site and landscape scales for wild buffalo (Syncerus caffer; Bar-David et al., 2009; Benhamou & Riotte-Lambert, 2012) and impala (Aepyceros melampus; Riotte-Lambert, Benhamou & Chamaille-Jammes, 2013) but finer-scale recursion to individual plants has not been investigated. Nevertheless, recursion has been explored amongst nectivorous insects and birds where the reuse of individual plants was found to occur after nectar replenishment (Davies & Houston, 1981; Bell, 1990; Williams & Thomson, 1998). Recursion behaviour has also been described in frugivorous primates returning to the same trees for fruit (Garber, 1988; Garber & Jelinek, 2006; Erhart & Overdorff, 2008; Janmaat, Ban & Boesch, 2013; Porter & Garber, 2013). As for nectar and fruit feeders, plant recovery period is also expected to strongly influence the movements and recursion frequency (rate) of grazers and browsers amongst sites. Prior to this study, the expected correspondence between individual plant recovery and recursion by wild grazers or browsers as an explanation for site recursion has not been explored.

Studies of recursion have important implications for animal population and habitat management (Bar-David et al., 2009). Most evaluations of wild animal resource requirements and preferences are based largely on the premise that if animals use resources (e.g., sites or food species) in lower or higher proportion to their availability then this suggests that the resource is avoided or preferred, respectively (Johnson, 1980). This framework is most commonly applied in studies that occur over relatively short time-frames to provide an indicative ‘snap-shot’ of resource-use. However, in natural environments resources are not consistently available in distribution, proportion and density through time and animals may reuse some resources but not others. An uncommon species of food-plant, for example, may appear to be a minor or unimportant part of the diet at selected sites but might be the subject of repeated use such that recursion would indicate it is highly selected. Alternatively, a common species of food-plant may appear to be avoided until investigation of recursion reveals reuse. Thus, studies of recursion are necessary to elaborate on spatial variation in availability and selection when assessing food and habitat preferences.

Recursion patterns may also be a useful indicator of population relations with habitat. Large herbivores, like elephants, are a particularly interesting species in which to study recursion as they are ecosystem engineers—having complex, scale-dependent effects on habitat structure and vegetative community (Bond, 1993; Jones, Lawton & Shachak, 1996). A study of recursion at the individual plant scale can identify if elephants are potentially over-utilising and depleting resources by re-browsing plants before they have recovered, or if they are facilitating growth of preferred or bulk-food plants (Fornara & du Toit, 2007; Cromsigt & Kuijper, 2011). Thus, increases in rates of recursion that exceed plant recovery rates could indicate that a population exceeds habitat capacity and reveal how they are influencing vegetation community structure and composition. Alternatively, plant recovery rates that exceed recursion may be evidence of further capacity to support greater elephant densities. An understanding of recursion patterns, therefore, may augment evaluations of a habitat’s capacity to support elephant and vegetation dynamics on the landscape under elephant grazing and browsing regimes.

In a previous study, English et al. (2014a) showed patterns of elephant recursion to sites consistent with site quality and optimal foraging theory. In this study we aim to test the hypothesis that recursion by elephants to sites in tropical rainforest also involves recursion to individual plants and corresponds with their plant recovery. We predict that elephant recursion would coincide with plant recovery and help to explain the periodicity of site recursion observed previously.

Materials and Methods

Study site and focal species

The dominant landform of the Lower Kinabatangan region is the extensive floodplain and its swamps. Soils are predominantly alluvial and derived from sedimentary deposits often rich in magnesium. Beyond the floodplains, soils are derived from sedimentary rocks (Azmi, 1998). The Kinabatangan floodplain is characterized by a warm, wet and humid tropical climate. The larger temperature variations are diurnal rather than seasonal. Mean monthly temperatures range between 21 °C and 34 °C (Ancrenaz, Calaque & Lackman-Ancrenaz, 2004). The north-easterly monsoon brings high monthly rainfall from October to February, although rainfall is also common from March to September. Dry months, with mean monthly rainfall <60 mm, tend to occur at roughly 3-year intervals. The mean annual rainfall is 3,000 mm (Acres & Folland, 1975).

This study focused on the area between the villages of Abai and Batu Puteh (5°18′-N 5°42′-N, 117°54′-E 118°33′-E), which were the downriver and upriver limits of the Lower Kinabatangan Wildlife Sanctuary (LKWS) elephant population’s range (approximately 200 individuals). The study area (approximately 218 km2) contains seven sections, each section referred to as a ‘lot’, including 89 km2 of protected forest reserves (Estes et al., 2012). The elephant herds utilised their whole range throughout the year including use of privately owned forests and cultivated land, particularly oil palm plantations that were adjacent to and between forested areas. Elephants in LKWS are mostly restricted to the linear fragments of forest along the Kinabatangan River (Estes et al., 2012) (Fig. 1).

Figure 1 Map of study site.

The Lower Kinabatangan Wildlife Sanctuary, Sabah, Malaysia (English et al., 2014b). Adapted from Clouded Leopard Project, Sabah www.cloudedleopard.org.

Plant recursion

Fourteen 50 m transects were located where elephants had fed previously. Transects were >300 m apart. One transect was established per day. We tracked fresh elephant signs including footprints, dung and signs of feeding to establish the transect along the group’s feeding path. All plants showing signs of elephant feeding within 2 m either side of transect were marked and labelled with the date and a reference number. Samples of all plant species were collected for identification at the Sabah Forestry Department Herbarium (SAN), Sandakan. The growth and recovery of each plant after herbivory was measured each month from April to December 2011 or until the elephants re-browsed the plant’s new growth. The length of the plant stem prior to browsing or grazing was determined by measuring the length of stems of the same plant that were not eaten, or remnants of the eaten stem, as a surrogate reference of original stem length. If the plant died or the new growth was re-browsed by other herbivores, thus preventing measurements of regrowth, this was recorded. It was possible to differentiate between elephant feeding signs and other herbivore feeding signs, such as from bearded pig (Sus barbatus) and sambar Deer (Rusa unicolor), because of the other sign and spoor left in the area, such as dung, footprints, the way in which the plant was eaten and the height of the sign. Recorded GPS positions of two collared elephants from the two main herds in LKWS confirmed when the focal group returned to the site and transect within the month and the age of the feeding signs allowed approximation of whether this coincided with the time of the focal elephant herd’s visit. If food plants had been re-browsed by elephant, but the focal group had not returned within the month, this was not recorded as a recursion.

Plant physiognomy varies among species and between plants within a species. Regrowth measurements were taken on a selected new shoot closest to the growth node nearest the feeding sign, or from the plant base, depending on plant physiognomy and how it recovered (see Fig. 2 for typological examples). Measurements included new shoot growth in length and basal diameter, and a count of the number of new shoots produced each month. The approximate length and the basal diameter of the original feeding sign on each plant were compared to the length and basal diameter of the new growth when it was fed on again. If we returned to a plant and it had been fed on since the last measurement was taken, the growth measurements from the prior month were used for comparison. The same technique was used for the two grasses: Phragmites karka—a reed, and Dinochloa scabrida—a bamboo, as their structure is a main stem with new growth emerging from nodes along the main stem, or from the root system. Short grasses were not included due to difficulty in identifying feeding signs (i.e., the whole plant is often ingested) and measuring recovery growth related to feeding by elephants. Student’s T-tests were used to compare the lengths of the individual plant new growth with their lengths when first selected for feeding to determine any significant difference.

Figure 2 Typological examples of plant growth forms.

Examples of plants selected by elephants in LKWS showing plant growth forms and their recovery. White arrows indicate portions of the plant eaten by elephant and black arrows indicate recovery growth.

Plant recovery growth and recursion rate

The expected time required for each individual plant to recover was estimated by averaging the monthly growth in length (mm) of the plant, divided by its estimated length at the beginning of study which was based on what remained of the stem after elephant feeding. Other stems on the same plant were used as a surrogate reference. Based on this monthly growth average, we estimated how many months it would take for the individual plant to return to its previous length. The difference in the average recovery time (months) subtracted from the average recursion time (months) is shown for each growth form (grass, ginger, palm, liana and woody species) where the plant species are the replicates used to derive standard errors for each growth form.

Multi-model inference and selection

An information theoretic approach (linear mixed effects model) was applied to test the hypothesis for recursion time and plant recovery time. Individual plant recovery times and time to recursion were used in the statistical analysis. We predicted recursion would occur after individual plants had recovered to their pre-herbivory height. We evaluated the power of plant recovery to explain recursion in the absence of other a priori hypotheses by comparing a model of our hypothesis with models that included random effects for site (transect) and growth form. We described and evaluated models in the ‘lme4’ package in R (Bates et al., 2014). All plants browsed by elephants, including those plants that did not receive recursion but were located on transects that received recursion, were included in the analyses. We used maximum likelihood (MLE) to provide estimates of the model’s parameters because fixed effects were different between models.

Results

Individual food plants, recently eaten by elephants, were identified by following the herd that was allowed to select sites and plants without influence. As one would expect for a herbivore selecting from a diverse landscape and flora we identified many individuals of commonly eaten species but a larger number of species represented by a few individuals. Thus, our sampling is skewed towards a few commonly eaten plants with many other species being eaten little by elephants and sampled less.

We recorded a total of 182 plants from 30 species eaten by elephants over 14 transects. Eighty-six of these plants from 12 species were re-browsed, i.e., recursion to individual plants (Fig. 3A). Twenty-six plants died and did not recover after being partially eaten by elephants and were, therefore, not included in further analyses (Fig. 3B). Two transects were not returned to by the elephants (five plants each transect), resulting in 146 plants used for recursion analyses.

Figure 3 Plant recursion and plant mortality.

(A) The percentage of plants along transects re-browsed by elephants. Nu, number of plants of each plant group eaten by elephants at the first visit; Nr, number of plants of each plant group that were re-browsed. (B) Plant mortality within plant growth forms.

The time to recursion for each plant species varied across the nine months of sampling (Table 1). Four species including ginger: Costus speciousus, grasses: Dinochloa scabrida, Phragmites karka and liana: Spatholobus sp., had recovered to their previous size when they were re-browsed, whereas the remaining eight species were re-browsed before they had fully recovered (Figs. 4 and 5). Seventy-seven percent of re-browsed plants were grasses. A linear mixed-effects model found that the plant recovery time is a good predictor of time to recursion but this varies as a function of growth form (grass, ginger, palm, liana and woody) and differences between sites. A large amount of variation is unexplained by recovery time (Table 2).

Figure 4 Recursion to grasses.

Recursion to grasses showing the average length of the grass stem when initially fed on (shaded bar) and the average total length of new shoots per month until recursion occurred, for two grass species (A) Phragmites karka and (B) Dinochloa scabrida. Standard error bars represent ±1 standard deviation of the sample distribution. Recovery has occurred when the black bar is the same length as the white bar.

Figure 5 Time to recursion-Time to recovery.

Time to recursion (months) minus Time to recovery (months) averages for each plant growth form. Numbers above bars represent the number of species within each growth form that received recursion. A positive value on the y-axis means that recursion occurred faster than plant recovery and a negative value means recursion occurred before plant recovery.

Table 1 Elephant food plants and recursion.

Plant genus/species eaten by elephants and those re-browsed at recursion during the 9-month study period. The number of plants eaten, number returned to, average time to plant species recovery and time to re-browsing at recursion are shown.

Growth form	Family	Genus/species	Plant eaten	Plant recursion	Average recovery time range (months)	Time to re-browsing (months)	
Grass	Poaceae	Phragmites karka	50	50	4–5	5	
Grass	Poaceae	Dinochloa scabrida	17	17	2–3	3	
Ginger	Costaceae	Costus speciousus	7	3	2–3	5	
Ginger	Maranthaceae	Donax canniformis	19	4	8–9	7	
Ginger	Zingiberaceae	Alpinia ligulata	8	0	8–11	a	
Palm	Arecaceae	Calamus caesius	3	1	8–9	4	
Palm	Arecaceae	Arenga sp.	4	1	7–9	4	
Palm	Arecaceae	Daemonorops sp.	3	0	11–14	a	
Palm	Arecaceae	Licuala sp.	3	0	10–15	a	
Liana	Leguminosae	Spatholobus sp.	3	1	3–4	4	
Liana	Leguminosae	Fordia sp.	3	2	8–12	2	
Woody	Guttiferae	Garciniaparvifolia	5	1	12–20	3	
Woody	Euphorbiaceae	Claoxylon sp.	2	2	9–16	5	
Woody	Dilleniaceae	Dillenia sp.	3	0	10–15	a	
Woody	Cornaceae	Alangium sp.	2	0	2–4	a	
Woody	Sapindaceae	Lepisanthes sp.	4	3	12–14	2	
Woody	Melastomataceae	Memecylon panniculum	2	1	5–6	2	
Woody	Myrtaceae	Szygium sp.	2	0	12–14	a	
Woody	Rubiaceae	Gardenia elata	2	0	10–12	a	
Woody	Hypericaceae	Cratoxylum sp.	1	0	11	a	
Woody	Phyllanthaceae	Bridelia sp.	1	0	12	a	
Woody	Euphorbiaceae	Mallotus sp.	1	0	14	a	
Woody	Rutaceae	Clausena excavata	1	0	2	a	
Woody	Euphorbiaceae	Macaranga sp.	2	0	6–8	a	
Woody	Euphorbiaceae	Paracroton sp.	2	0	12–15	a	
Woody	Meliaceae	Dysoxylum sp.	1	0	3	a	
Woody	Lamiaceae	Callicarpa sp.	2	0	5–7	a	
Woody	Leeaceae	Indica sp.	1	0	12	a	
Woody	Phyllanthaceae	Antidesma thwaites	1	0	16	a	
Woody	Apocynaceae	Rauvolfia sp.	1	0	14	a	
Notes.

a Represents plant species that did not receive recursion.

Table 2 Multimodel inference and selection.

Four models ranked in order of AIC weights where recovery time (months) is the fixed-effect and plant growth form and site are random-effects, the response variable is time to recursion (months).

Model	Fixed effects	Random effects	n	K	AICc	Δ AIC	ω	
1	Recovery time	Growth form & site ID	146	4	561.9	0	0.6300	
2	(Base model)	Growth form & site ID	146	3	563.2	1.3	0.3300	
3	Recovery time	Growth form	146	3	567.1	5.2	0.0500	
4	Recovery time	Site ID	146	3	612.1	50.2	0.0000	

Two species of grass, Phragmites karka—a reed, and Dinochloa scabrida—a bamboo, received recursion at a time when their length was not significantly different from when they had first been selected (Reed: t-test , df = 50, P = 0.137; Bamboo: t-test , df = 17, P = 0.232) (Fig. 4). However all other growth forms that were selected were re-browsed before they had recovered to their previous length except one ginger species, Costus speciousus, and one liana species, Spatholobus sp. Mean recursion time subtracted from mean recovery time to all growth forms illustrates that recursion to palms, lianas and woody species occurred many months before the individual plants had recovered (Table 1 and Fig. 5).

Discussion

Plant recovery time after herbivory as an explanation for site and plant recursion is expected from optimal foraging theory and has been postulated for large wild herbivores (e.g., Bar-David et al., 2009) but had not yet been investigated in uncontrolled environments. We found plant recovery time to be a good predictor of time to plant recursion by elephant but also observed large variation with differences in growth form and amongst sites. The recovery time of the two primary species of grass—elephants’ primary food species—coincided with plant and site recursion but this was not also true for browsed species. Palms, lianas and woody species were re-browsed before they had recovered. The large amount of variation attributable to sites may pertain to a number of abiotic influences on plant recovery rates, such as soil fertility and micro-climate.

Recursion rate corresponded best with the recovery of grasses: Dinochloa scabrida and Phragmites karka, even though those grass species had very different recovery times: e.g., 2–3 and 4–5 months, respectively. In all cases these grazed plants were returned to after they had recovered. Grasses have also been identified as the preferred food plants of the elephants (English et al., 2014b) and they made up 43% of all plants selected and 77% of plants receiving recursion. Elephants are bulk feeders so it is likely that they time their returns to bulk feed on grasses and grassed sites when stands have recovered sufficiently to meet their intake requirements. Grasses were less likely to die and faster to recover compared to other growth forms. Other than grasses, most species did not receive recursion, or if they did, it occurred before the individual plants had recovered.

Recursion to a few poorly regenerated species, specifically woody trees, palms, lianas and one species of ginger, may be a result of elephants foraging on other nearby plants (i.e., grasses) and indiscriminately re-browsing those unrecovered plants. If this was the case we would expect the re-browsing of plants prior to their recovery would be most common when they are found within grass-dominated sites. Half of the browse plant samples found within grass-dominated sites were re-browsed before their recovery compared to 20% of those outside grassed areas. Thus, premature woody-plant re-browsing could be an ancillary to grazing sites.

Alternatively, recursion to unrecovered plants may be due to elephants specifically targeting those growth forms or their younger growth because they contribute a small but important component of the diet (e.g., trace elements). Forest plant productivity and nutritional quality has been found to be highest after around 5–6 months of plant regrowth (Plumptre, 1993), which might explain why some browsed plants were re-eaten after this many months of regrowth but before the recovery of their branch and stem lengths. Furthermore, elephants may select some food plants not just to facilitate re-growth productivity but also to manipulate the structure and composition of the plant community at sites. For example, Jachman & Bell (1985) proposed that African elephants selectively fell preferred tree species to stimulate coppicing but also to increase the availability of other palatable forage species. Elephants may, therefore, alter structure and floristic composition, especially of woody species, in ways that increase rather than reduce carrying capacity. If woody plants are re-browsed faster than they can recover then elephant feeding might lead to the creation and maintenance of open, grassed areas. These areas are likely to become dominated by early successional species, thus providing the elephants with more of their preferred food such as grasses. Therefore, feeding on woody species faster than they can recover may augment grass patches and prevent woody invasion. This is a common observation of elephants. As ecosystem engineers they are known to alter the structure and composition of habitat and plant communities (Laws, 1970; Bryant, 1981; Bergström & Danell, 1987; du Toit, Bryant & Frisby, 1990; Ben-Shahar, 1993; Prins et al., 1998). Elephant impact on woody vegetation has led to decreasing numbers of trees and increase of open areas in Africa (Conybeare, 2004; O’Connor, Goodman & Clegg, 2007). The results of our study suggest that elephants in LKWS may be controlling re-forestation within open grass areas by re-browsing on woody species, lianas, and palms before their recovery. However, a long-term study on elephants as ecosystem engineers within the Lower Kinabatangan is required.

Another plausible explanation for recursion on poorly regenerated plants is that these plants are highly desirable and resources in the area are inadequate, perhaps due to spatial constraints, habitat fragmentation and overstocking. Resources may be insufficient to support a slower site recursion rate. Elephant feeding on plants before they have recovered might indicate that food species are being over-exploited and that the elephant population is approaching or exceeding habitat carrying capacity. However, with the exception of grasses, our results show that only 19 of 89 browse plants were returned to for feeding during the nine-month study period. This finding suggests that there is no evidence from recursion data that this elephant population has exceeded the area’s carrying capacity. The two first explanations, individually or in combination, best explain the pattern.

Limitations in data collection in this study are imposed by the lack of independence between plant samples within species and amongst sites that were determined by elephant movements and choices and also due to plant distributions within the study area, particularly for Poaceae, which occur in homogenous stands at just a few sites that are highly favoured by the elephant (English et al., 2014b). Sites and transects are not balanced replicates for each plant species measured, and recursion occurred to most but not all selected sites. A lack of equal distribution of all species across all transects due to elephant food plant choices and plant species heterogeneity and distribution influenced the strength of the data. Moreover, despite a satisfactory sample size of browsed plants initially, a lack of re-browsing to plant samples across a variety of species resulted in reduced sample sizes of plants receiving recursion, especially for woody species, and therefore limited statistical power.

Despite these limitations, we established the likely importance of recovery time for recursion of elephants’ bulk food, grass. It was a novel approach to establish the relationship between resource recovery and recursion by elephants in LKWS by measuring plant recovery rates in an uncontrolled environment. Our results recognised the importance of incorporating open land, for elephants to feed on grasses, into corridor design and reforestation programmes in the area. Future studies investigating recursion to plants could be improved by ensuring a relatively even distribution of plant samples across all transects and increasing sample sizes of each species. For example, by establishing more transects or extending sampling distance (>2 m either side of transect) in order to incorporate a larger number of samples within each species for statistical comparison. It would also be beneficial to compare inter-annual variation in re-browsing and elephant impact on their resources as ecosystem engineers.

Supplemental Information

Supplemental Information 1 Raw Data Recursion

Click here for additional data file.

Additional Information and Declarations

Competing Interests

Author Contributions

Animal Ethics

Field Study Permissions

The authors declare there are no competing interests.

Megan English conceived and designed the experiments, performed the experiments, analyzed the data, contributed reagents/materials/analysis tools, wrote the paper, prepared figures and/or tables, reviewed drafts of the paper.

Graeme Gillespie conceived and designed the experiments, contributed reagents/materials/analysis tools, prepared figures and/or tables, reviewed drafts of the paper.

Benoit Goossens and Marc Ancrenaz contributed reagents/materials/analysis tools, reviewed drafts of the paper.

Sulaiman Ismail performed the experiments, contributed reagents/materials/analysis tools.

Wayne Linklater conceived and designed the experiments, analyzed the data, contributed reagents/materials/analysis tools, wrote the paper, prepared figures and/or tables, reviewed drafts of the paper.

The following information was supplied relating to ethical approvals (i.e., approving body and any reference numbers):

Victoria University of Wellington Animal Ethics Committee approved the research on the 30/3/2010. Elephants were not manipulated in any way for the purposes of this research. GPS tracking collars were attached to elephants for another student’s research supervised by Dr. Benoit Goossens, who allowed Megan English to access the data information.

The following information was supplied relating to field study approvals (i.e., approving body and any reference numbers):

Economic Planning Unit, Kuala Lumpur and Sabah Wildlife Department. Research pass number: 2833.

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
