# Peer review of "Recursion to food plants by free-ranging Bornean elephant"

_PeerJ, doi:10.7717/peerj.1030_

## Round 0.1 · original submission · Major Revisions

Both reviewers have identified substantive problems with the paper. Reviewer 1 is concerned about the low number of species individual of many species that were tagged in the initial survey. Why was this the case? Perhaps there were low numbers of these species encountered along the transects? Please explain. Also, given the low sample sizes for many species, it makes more sense sense to use growth form instead of species in the statistical model.

Reviewer 2 offers valuable advice on redrafting parts of the manuscript. Please follow this reviewer's advice in clarifying the unique contribution made by this paper relative to cognate ones published by the same authors.

Reviewer 1 ·

Basic reporting

Well written.
Good logical flow.
Appropriate length for content.
Well structured.
Tables and figures are all necessary.
Paper is self contained.

Experimental design

Approach to the study and the general features of the design were good.
Scope is within the ambit of the journal.
The study addresses a clearly defined research question.
Methods are adequately described.
There are no obvious ethical considerations.
My issue is with sample size rather than with design.

Validity of the findings

The findings about the two grass species appear robust, but the overall conclusion is not convincing on account of the gross disparity in sampling intensity across species. Concerns are detailed in "General Comments for the Author".

Additional comments

This study addresses the question of whether the rate of regrowth of a plant following defoliation influences the time taken for recursion by the original herbivore species, using elephant in a Bornean forest as the study species. This is a useful question from the perspective of both herbivore impacts on plants and the foraging ecology of mammals. It is not a well explored question and the study therefore potentially makes a useful contribution. It should be of broad interest.

The overall design of the study appears sound for a general statement about foraging impacts of these elephants. Areas in which elephants had foraged were sampled in an adequately rigorous manner (14 50 m transects), and the regrowth of plants was followed for a sufficiently long period (nine months) and measured at a reasonable temporal resolution (monthly).

Approach to analysis also appears sound in terms of the nature of statistical treatment undertaken. However, I was disappointed by a sample size of only 146 plants, especially considering it was spread across 30 species. This is a very small number of plants to measure in a two week period (duration of the baseline exercise) – little more than ten plants a day. Less than a handful of individuals were sampled for most species, yet species were an explanatory factor entered into the analytical models. The study would have been more robust had a concerted effort been made to locate at first sampling a decent sample size of individuals for each species, perhaps sacrificing the number of species which were followed. Concentration of sampling on two grass species has also distorted results. It is not only individual species which have been compromised by the small sample size – that of lianas and palms as collective growth forms is each very small. This shortcoming is also evident in the lack of error bars for a number of species in Figure 5.

An additional comment on the statistical analysis is that “Time until recovery and time until recursion were averages of plants”, that is they used a mean as a fixed effect but for which there is an associated measure of variance, without recognising the consequences for statistical procedures.

The study reveals that recursion to grasses allowed regrowth to size at time of first defoliation, whereas the same was not observed for most species of other life forms. This is a useful finding but it would have been stronger had there been a priori recognition of the need to spread sampling across species and growth types more equitably than it was. The fact that many species experienced recursion before they had regrown to previous size would seem a contradiction of the statement “plant recovery time strongly influenced the time to recursion” (lines 188-190). I would argue that a universal statement of this nature should be evident across growth forms and species and should not be a consequence of the dominance of the statistical analysis by two species of the same growth form.

Minor points
Line 44 word appears to be missing between ‘scales’ and ‘wild buffalo’
Lines 50-1 awkward phrasing
Lines 96-98 should this not appear in acknowledgements
Line 133 ‘between’ = ‘among’.
For figures 4 and 5, it needs to be confirmed if the bars are SD, SE or confidence intervals. For figure 5, why do B i, ii, and iii; C I and ii, and D i not have bars?
A short statement about climate and soil fertility (geology could be a surrogate) would be useful for interpreting potential abiotic influences on regrowth rates. Is it a seasonal environment, which would influence the rate of regrowth over the year?

Reviewer 2 ·

Basic reporting

Overall, I think that exploring how recovery time might influence how elephants use/reuse resources is very interesting and sheds new light on herbivore-plant interactions. However, the way the paper has been written unfortunately clouds the study’s findings, not displaying them to their full potential. In addition to the results not being conveyed clearly, the Introduction and Discussion seem to have sections that are tangential to the core aim of the study, which detracts from the study’s exciting findings. Please rework the manuscript such that it only includes topics or information that support the key aim of the study (“We aim to test if recovery time contributes to explaining species and site recursion patterns by testing the hypothesis that recursion to plants by elephants in tropical rainforest is influenced by plant recovery rates”).

One major point of potential confusion is the author’s use of the word “plant” throughout the manuscript. The term is quite ambiguous because “plant” could refer to an individual plant or it could be used to indicate a plant species. Please indicate which meaning is intended throughout the manuscript.

The introduction is very repetitious. For example, see the first and second paragraphs: they both essentially touch on the same topics, but the second paragraph includes the references. The authors should streamline this by combining these two paragraphs.

I think it would strengthen the manuscript if all of the results are stated more clearly and refer back to the study's overall aim in the first paragraph of the Discussion. This way, the reader is clued in to the key findings and the authors can then use that first paragraph of filled with the main findings as a starting point for the remainder of the discussion. At the moment, the findings are not linked clearly with the aims of the study. With a bit of tweaking and re-writing, this manuscript would be much stronger and have a larger impact.

Minor comments regarding the basic reporting:
I also found word choice to be repetitious throughout this manuscript (for example L114-116, the word “establish(ed)” is used 3 times). Please read through the manuscript and change over-used words.
Please change the term “feed sign” to “feeding signs” because “feed sign” comes across as very colloquial.

Figures are referenced in several different ways throughout this manuscript (e.g. L184 “(Fig 4&5), L187 (Figure 5)). Be consistent.

Please be consistent in how you refer to plant species. I noticed sometimes the common name for a family and sometimes the scientific name is used. Be consistent. For Figure 5, why are the common group names listed, and then the scientific names used to label each graph? It would be easier for the reader if you listed the common group name (e.g. gingers) with the scientific species names next to them so that the legend would read “Recursion to plant species. The average length of the stem when initially eaten (white bar) and average monthly total new shoot length of each species until re-browsing A) Gingers: i) Costus speciousus and ii) Donax canniformis…”

L124-127. Please rephrase this sentence because it is not written clearly.

Figure 2 is not of sufficient resolution.

Experimental design

The aim of this study was to “ test if recovery time contributes to explaining species and site recursion patterns by testing the hypothesis that recursion to plants by elephants in tropical rainforest is influenced by plant recovery rates,” however, this manuscript did not really focus on site recursion patterns. I notice these authors also published an article entitled “Foraging site recursion by forest elephants” which the above aim also seems to include. Why were these studies not written as a single unit? It could have been laid out similarly to Shrader et al. 2012 “Forest or the trees: At what scale do elephants make foraging decisions”.

Because they were not written as a single story and, the other manuscript focusing on site recursion is already published, I think it is extremely important to focus on different points for this manuscript. Thus I would not focus on “site” here, instead focus on the finer scale (individual plant recursion). Please make the appropriate changes to ensure that this paper is unique.

In general, the sampling methods are clear. However, I find it very hard to comment on the statistical models because they are not explained well. Much later (in the results) the authors mention some of the statistical models used (linear fixed-effects model, t-tests). All statistical models used must be explained in the methods with reasoning about why you used them. Currently, this is not the case. Please add in explanations and support for the statistical models used.

Minor comments:

How many elephants live in the study area? Add to Methods.

In Table 1, please remove the sentence about where the species were identified. That information was already noted in the Methods section. Also, please fill in values (or a symbol) for the blank spaces in the “time to rebrowsing” column. For species never rebrowsed, note that in that column.

Validity of the findings

As I have mentioned above, the results and conclusions of the study are not as clearly stated as one would hope. I think that the overall findings are very interesting and novel, however, I think that they need to be displayed more clearly. If the authors clarify the above points I think this manuscript would be much stronger.

Additional comments

No Comments

---

## Round 0.2 · accepted · Accept

The revised ms has incorporated the concerns of both reviewers. The inclusion of text on the sampling strategy has alleviated concerns about abundances of samples among species. The clarity of the text is greatly improved.